# Sensing Senses: Optical Biosensors to Study Gustation

**DOI:** 10.3390/s20071811

**Published:** 2020-03-25

**Authors:** Elena von Molitor, Katja Riedel, Mathias Hafner, Rüdiger Rudolf, Tiziana Cesetti

**Affiliations:** 1Institute of Molecular and Cell Biology, Hochschule Mannheim, 68163 Mannheim, Germanyt.cesetti@hs-mannheim.de (T.C.); 2BRAIN AG, 64673 Zwingenberg, Germany; kar@brain-biotech.com; 3Interdisciplinary Center for Neurosciences, Heidelberg University, 69120 Heidelberg, Germany

**Keywords:** taste signaling, calcium, imaging, optical biosensors, tastants, gustation

## Abstract

The five basic taste modalities, sweet, bitter, umami, salty and sour induce changes of Ca^2+^ levels, pH and/or membrane potential in taste cells of the tongue and/or in neurons that convey and decode gustatory signals to the brain. Optical biosensors, which can be either synthetic dyes or genetically encoded proteins whose fluorescence spectra depend on levels of Ca^2+^, pH or membrane potential, have been used in primary cells/tissues or in recombinant systems to study taste-related intra- and intercellular signaling mechanisms or to discover new ligands. Taste-evoked responses were measured by microscopy achieving high spatial and temporal resolution, while plate readers were employed for higher throughput screening. Here, these approaches making use of fluorescent optical biosensors to investigate specific taste-related questions or to screen new agonists/antagonists for the different taste modalities were reviewed systematically. Furthermore, in the context of recent developments in genetically encoded sensors, 3D cultures and imaging technologies, we propose new feasible approaches for studying taste physiology and for compound screening.

## 1. Introduction

On the journey towards understanding the sense of taste, two major parameters have been measured at the cellular level: voltage and Ca^2+^ changes. Indeed, both types of signals are emitted by taste cells in the periphery and by neurons in the central nervous system in response to a given tastant. All the way up from the tongue to the brain, our body uses an alphabet of voltage and Ca^2+^ fluctuations to decipher qualities and concentrations of tastants and to instruct decisions on food palatability. These notions have been used to construct and use different types of biosensors for fundamental and applied research.

Biosensors are analytical devices that allow the detection of chemical analytes through the combination of a biological recognition element with a physicochemical transducer [1,2]. Amongst these, cell-based biosensors incorporate sensors (and sometimes also transducers) in living cells and convert the resulting information into digital signals, which are then analyzed by means of specific techniques [2,3]. With respect to taste analysis, cell-based biosensors are also called “bioelectronic tongues” (B-ETs). These are able to analyze tastants in a fast, sensitive and selective way and have been recently designed based on the discoveries in taste physiology [2,4,5]. Since B-ETs are based on cells or tissues expressing taste receptors, they provide specificity, selectivity and sensitivity analogous to the biological sense of taste [5]. So far, transducers coupled to B-ETs use often voltametric/amperometric, potentiometric or piezoelectric methods and thus include microelectrode arrays, field effect transistors or light-addressable potentiometric sensors [5,6,7,8]. Even if these prototype devices have not yet been commercialized, B-ETs have a wide range of potential applications in the fields of drug and food industries, toxicology and biomedicine [3,9,10]. Furthermore, they might become valuable tools for research on taste transduction mechanisms. Finally, their development is tightly linked to the progress in understanding taste physiology.

A second kind of approach that has been primarily applied to basic research on taste sensation uses “molecular optical biosensors” to measure cellular responses to flavor stimulation. In this context, we refer to molecular optical biosensors as molecules that produce light changes in relation to cellular activity, therefore they are suitable to measure functional cell responses. They can be either, chemical fluorophores or genetically encoded fluorescent proteins, that shift their excitation/emission spectra or fluorescence intensity based on specific biological signals, such as changes in ion concentration, voltage, metabolites and second messenger molecules. Molecular optical biosensors are highly sensitive, real-time and simple-to-operate tools, that can be easily monitored with a plate reader for enhanced throughput or with microscopy for high spatial resolution. In contrast to classical B-ETs, molecular optical biosensors have been essential in basic research in the field of taste, but they have not yet been considered for the applications typical for B-ETs. This contribution reviews the use of molecular optical biosensors as tools in taste research and proposes their integration into the concept of B-ETs. Starting with a summary of the current knowledge on the taste system, the manuscript will systematically address applications of different molecular optical biosensors in the tongue, the central nervous system and in recombinant cell systems. Then, we will speculate on a future use of molecular optical biosensors to novel fields of applied taste research.

## 2. Taste—Function and Mechanism of Action

The taste system is crucial for feeding and survival as well as to detect and discriminate a large amount of chemical substances in complex environments. The sensation of the five basic taste modalities sweet, bitter, umami, sour and salty, gives essential information on food choice and prepares the body for digestion. Each taste modality has a specific role in body homeostasis and taste receptors (TRs) are also expressed in extraoral tissues, where they mediate non-tasting effects, e.g., in the airway, gastrointestinal tract or heart (reviewed in [11,12,13]). In humans, the sweet taste is the most attractive modality, probably because sugars are associated with calories and energy fuel for the body. Umami, the taste of glutamate, reflects the protein content in food while bitter substances are perceived as unpleasant, which likely reflects a warning signal for toxicity. Salty and acid tastes can be attractive or repulsive, depending on their concentration which is critical for mediating body-fluid maintenance and the acid-base balance [14,15]. For the modalities sweet, umami and bitter, flavors are recognized via highly specialized TRs, while salt and acid sensations utilize the activation of ion channels. From a clinical point of view, understanding the functionality and the complex signaling networks downstream of these receptors and ion channels is a major goal, to treat e.g., obesity, diabetes or cardiovascular dysfunctions. On another note, the development of bitter inhibitors may help to improve palatability of unpleasant bitter medicines. For example, special formulations for children are often sweetened although this still does not completely eliminate the bitterness of many drugs [16,17]. Finally, food industries are highly interested in understanding taste physiology to optimize their products with new flavors or substitutes [18,19].

### 2.1. Taste System Anatomy and Transduction in the Periphery

Taste buds are the peripheral organs of gustation. They are implanted in the epithelia of the tongue, palate and epiglottis [20]. In human beings, ~5000 taste buds are distributed on the tongue and are organized in taste papillae (Figure 1) [21,22]. Fungiform papillae are found in the anterior two thirds of the tongue, they contain ~3.5 taste buds each and are innervated by the chorda tympani nerve. Foliate papillae reside on the posterior lateral sides of the tongue. They contain multiple taste buds and are innervated by both, the corda tympani and the cranial nerve IX, which belongs to the glossopharyngeal nerve [22,23]. Circumvallate papillae on the dorsal tongue are the largest papillae with ~250 taste buds. They are only present in mammals [24] and are innervated by the glossopharyngeal nerve [25]. 

Each taste bud contains 80–100 mature polarized neuroepithelial taste cells of elongated shape. They are grouped into three categories (Figure 1) [15,20,26]. Type I cells are the most abundant cells and believed to have a glia-like function. They synthesize NTPDase to degrade ATP released by other cells for intercellular communication [15,26,27,28]. Type I cells show an electron-dense cytoplasm, apical dark granules and long microvilli. Conversely, type II cells contain a more electron-lucent cytoplasm and their microvilli are shorter and thicker [29,30]. They are also called “receptor cells” as they express G-protein coupled TRs, which are key players in the transduction of sweet, bitter and umami taste. In addition, type II cells express voltage-gated Na^+^ and K^+^ channels to mediate action potentials [15]. Upon depolarization, hemichannels in type II cells allow the release of ATP to stimulate nerve terminals and type III cells [31,32,33,34,35]. The latter are also termed “presynaptic cells” as they synapse with afferent nerve terminals and release serotonin [36,37,38] as well as other neurotransmitters [39,40,41,42]. Furthermore, type III cells are involved in sour taste transduction [43]. They display an intermediate electron density and a single microvillus [29,30]. As the life span of all taste cells is limited to only ~10 days, they need to be replenished regularly. To this end, progenitor cells, also called type IV cells, reside on the basement of the taste bud and give constantly rise to new mature taste cells [15,26,44].

#### 2.1.1. Taste Transduction of Saltiness

Saltiness is associated with the taste of table salt (NaCl) and Na^+^ ions [45]. At low concentrations, salt is attractive, while at higher concentrations it is aversive [46]. Apparently, different taste cell types and mechanisms are responsible for these responses. The attractive pathway is selective and sensitive to low concentrations of Na^+^ (10 mM–150 mM) and mediated by amiloride-sensitive Na^+^ channels (ENaC) [47,48]. The exact channel composition, the downstream signaling, and the cells expressing ENaC are still under debate [49,50,51]. The amiloride-sensitive pathway plays only a minor role in taste perception in humans, with large variability among subjects [51,52]. The amiloride-insensitive aversive pathway responds to a wide range of salts. It is activated only by higher salt concentrations and mediated by a different but still undefined cell population (Figure 2) [46]. Two mechanisms for amiloride-insensitive salt sensation were proposed. First, the vanilloid receptor (V1R) is a candidate [53,54], since mice lacking V1R showed altered nerve responses to salt [53]. Second, an unknown Cl^−^ conductance was found to mediate a Ca^2+^ rise in type II cells in response to high salt concentrations [55]. 

#### 2.1.2. Taste Transduction of Sour

Sour taste mediates both attractive and aversive behaviors depending on tastant concentration, context (for example presence of sugars) and learning. It is mediated either by organic or mineral acids. Organic acids are sensed by presynaptic type III cells upon changes in intracellular pH [56,57] which induces serotonin release [56]. Such weak acids might permeate the cell membrane in the protonated form [58], dissociate in the cytoplasm and protons may then block Kir2.1 channels, resulting in depolarization and voltage-gated Ca^2+^ entry [59]. For the sensation of mineral strong acids, a variety of mechanisms have been discussed (Figure 2) [60,61,62,63,64,65] (reviewed in [25,58]). One candidate was the polycystic-kidney-disease-like ion channel (PKD2L1). However, while selective ablation of PKDL1 expressing cells eliminated gustatory neural response [66,67], sour sensation remained intact in PKD2L1 knockout mice [68]. Recently, a new mechanism has been proposed: extracellular protons derived from the dissociation of strong mineral acids might enter type III cells via Otop1, a specific ion channel now termed “sour receptor” (Figure 2). However, despite a large reduction in nerve response to strong acids, Otop1 knockout mice have a retained acid aversion [69]. Therefore, the mechanism leading to the acid-induced avoidance behavior remains elusive, and further transduction pathways may be involved. Furthermore, sour taste may utilize different pathways depending on the species [60,61,62,63].

#### 2.1.3. Taste Transduction of Bitter

In humans, bitter taste is mediated via ~25 bitter taste receptors (T2Rs), which can be co-expressed in different combinations in type II cells [70,71]. Many structurally unrelated compounds are able to activate T2Rs, and while some bitter compounds are specific for one receptor, others can activate more than one [47,72,73]. Studies on native rat receptor cells suggest cellular specificity for certain bitter compounds [74]. T2Rs are G-protein coupled receptors (GPCR) [71,75], mainly linked to a specific taste G-protein named “gustducin” (Figure 2). Even if other G-proteins may be co-expressed in receptor cells, gustducin is the main G-protein involved in bitter transduction [76,77,78]. Upon stimulation of T2Rs, the βγ-subunit of gustducin is believed to stimulate phospholipase Cβ2 (PLCβ2) to mediate Ca^2+^ release from intracellular stores via inositol-3-phosphate (IP3) [79]. Subsequently, cell depolarization is thought to be mediated by Ca^2+^-activated transient receptor potential M5 channels (TRPM5) [80], which leads to ATP release via pannexin or CAHLM channels followed by stimulation of afferent fibers and presynaptic type III cells [25,81,82]. In addition, the α-subunit of gustducin was found to activate a phosphodiesterase (PDE) to keep intracellular cAMP levels low and to avoid the activation of protein kinase A, which in turn inhibits PLCβ2 [43].

#### 2.1.4. Taste Transduction of Sweet

Sweet taste is mediated by caloric sugars and artificial sweeteners upon binding to the sweet receptor which is composed of the T1R2-T1R3 heterodimer (Figure 2) [83,84,85]. Receptor cells sensing sweet and bitter compounds are segregated in two different populations [22]. Sweet-sensing cells use the classical bitter downstream signaling involving PLCβ2, IP3, Ca^2+^ release, TRPM5 and ATP release (Figure 2) [43,83]. However, there might be an alternative transduction pathway for sweet, as T1R3 knockout mice showed residual responsiveness to sugars, but not to artificial sweeteners [86]. This may be mediated by glucose transporters (GLUT and SGLT) and downstream K_ATP_ channels [87]. Additionally, before the discovery of the sweet taste receptor, it was proposed that sugars activate adenyl-cyclase (AC) to increase cAMP levels, favoring protein kinase A (PKA) mediated inhibition of K^+^ conductance resulting in cell depolarization (Figure 2) [43,88,89,90,91,92]. It is discussed, that this pathway is mostly utilized by sugars while artificial sweeteners may largely activate the classical PLCβ2 / IP3 pathway [43].

#### 2.1.5. Taste Transduction of Umami and Additional Taste Qualities

Umami is associated with L-glutamate and guaninmonophosphate/inosinmonophosphate which are produced after hydrolysis of proteins and NTPs [15,45]. As sweet, also umami is sensed by type II cells via T1R1-T1R3 heterodimers (Figure 2) [93,94]. The downstream signaling of umami taste is thought to be similar to that of bitter taste [95].

Carbonation has been recently recognized as an additional taste modality. In type III cells, CO_2_ is sensed by carbonic anhydrase 4 (CA4) which produces protons and bicarbonate ions. Proton influx into acid-sensing cells is believed to serve for signal transduction of carbonation [66,96].

Dietary lipids are perceived via GPR120 and CD36 receptors that activate a phospholipase A2 which was reported to trigger Ca^2+^ release from the store and Ca^2+^ entry via multiple store-operated Ca^2+^ channels such as Orai1 and 3 [97,98]. 

Kokumi, the sensation of mouthfullness, is produced by substances that have no taste on their own but enhance other modalities. It is responsible to inform the organism about the presence of amino acids and proteins. Accordingly, it is activated by small peptides, amino acids and Ca^2+^. Indeed, the calcium sensing receptor (CaSR) is believed to be responsible in this context. CaSR is present in type II and III cells, but not co-expressed with sweet receptors [99,100]. 

Capsaicin and vanillin evoke a burning sensation when ingested due to binding to V1R, located in the nerve terminals [101,102]. It is debated, whether a variant of V1R is also present in taste cells. V1R was shown to be responsible for capsaicin response in taste receptor cells and to contribute, via capacitative Ca^2+^ entry, to Ca^2+^ homeostasis together with mitochondria and Ca^2+^ exchangers [103]. Furthermore, it was also proposed to account for amiloride-insensitive salt responses in rat [53].

#### 2.1.6. Communication of Taste Cells

To stimulate the nerves and to communicate with each other, taste bud cells release different neurotransmitters and paracrine molecules. Type I cells may also release some neurotransmitters such as GABA [104], however one of their major roles is rather the clearance of ATP by released NTPDase (Figure 1) [26]. Type II cells stimulate the nerves by ATP release (Figure 2) [33,34]. Accordingly, nerve terminals express purinergic receptors (Figure 2) [31]. In addition, ATP also stimulates presynaptic type III cells and exerts a positive autocrine feedback on type II cells (Figure 1) [105], which express M3 muscarinic receptors and release acetylcholine to generate positive autocrine and paracrine signals [42,106,107,108]. Type III cells make synaptic contacts with afferent fibers and release serotonin (Figure 2) [36], noradrenaline [39] and GABA [40]. While Serotonin and GABA act not only on nerve terminals, but have also inhibitory functions on type II cells, the role of noradrenaline is not well understood [43]. Molecules involved in the different signaling pathways can be used as markers to identify the different cell types in a taste bud. Thus, type I cells are recognized by expression of NTPDase [109], and GLAST [110], type II cells by gustducin [79,111], TRPM5 [112,113], IP3R [50] and PLCβ2 [33,49], while type III cells mainly express GAD1 [42,114] or SNAP25 [49]. Additionally, it is possible to discriminate between type II and type III cells according to their response to KCl-mediated depolarization, which induces Ca^2+^ entry via voltage-gated Ca^2+^ channels only in type III cells [49,50].

### 2.2. In Search of an Appropriate System to Study Taste Signaling with Biosensors

Much progress in our knowledge about taste has been achieved in the last decades. However, major obstacles have still not been solved and hamper further research. The main challenge is the absence of a reliable, functionally complete test system to study human taste (reviewed in [24]). So far, taste has been predominantly studied in animals due to limited accessibility to human samples. However, interspecies differences have been widely recognized. For example, taste nerve activity results in different outcomes for carnivores and herbivores when stimulated with salts such as NaCl or KCl [115]. Additionally, the number of TRs is variable among species, e.g., while frogs express 49 bitter receptors (T2R), the numbers for chicken, dogs, cows and humans are 3, 15, 18 and 25, respectively [116,117,118]. Other species, such as domestic cats, lack sweet taste receptors altogether and thus are not attracted by sweetness [119]. Even the shape of taste buds differs: while humans contain oval taste buds, they are spindle-like in pigs or melon-shaped in horses [24]. Most of the research on taste has been performed on rodents. However, the sweet receptor, which was identified first in rat [83] and then in humans [93], displays structural and functional differences between the two species, as the selective competitive inhibitor lactisol and the artificial sweetener aspartame are effective only in humans [120,121]. Furthermore, while rodents are strongly attracted by polysaccharides derived from starch, humans are not [122,123].

An alternative to primary rodent tissue is the use of cell lines expressing heterologous functional gustatory receptors together with molecules essential for the signal transduction (Figure 3). These recombinant expression systems can be used for high throughput screenings of a large number of chemicals but sometimes the results might not be representative of the situation in vivo, as they lack the complexity of native tissues. To study human taste, it would be best to have access to human samples. In principle, this is possible, since papillae can be donated as they regenerate [124]. However, even when donors were found, papillae are naturally composed of mixed cell populations, which have a limited life span. Indeed, up to now primary human taste cells can be maximally cultured for seven passages [125] and heterogeneity and variability of the samples are critical. To overcome these limitations, immortalization of human taste cells yielded a test system able to reliably sense bitter substances [117]. Alternatively, isolated and immortalized progenitor taste cells could be used to differentiate into mature taste cells or to build organoids [126,127]. However, the knowledge on how to differentiate these progenitors is still at the beginning [128] and nobody has so far attempted to isolate, cultivate and differentiate human progenitors cells. The different biological models that have been used and analyzed with optical biosensors to progress in the study of taste physiology up to now can be grouped into four main categories as follows (Figure 3):

Isolated animal primary taste cells, taste buds, tongue epithelia and slices were used in combination with fluorescent dyes in ex vivo live imaging experiments for unravelling the intracellular signal transduction pathways and intercellular communication.Recombinant systems expressing taste receptors and downstream signaling molecules in non-taste cell lines were employed upon loading with chemical dyes in plate reader experiments to study receptor structure, binding sites, selectivity and sensitivity in high throughput.Biosensor cells expressing specific neurotransmitter/hormone receptors were used upon loading with fluorescent dyes and juxtaposed to taste cells/tissue to monitor with live imaging experiments the release of neurotransmitters such as ATP, serotonin, noradrenaline, GABA and acetylcholine.Expression of genetically encoded biosensors in neurons of mice to monitor brain activity patterns in response to flavor application in vivo and to label specific cell types in a reporter gene mode.

## 3. Use of Molecular Optical Biosensors for Taste Research

In the following paragraphs, the application of different molecular optical biosensors for the study of taste sensation will be systematically reviewed according to their use in taste buds and the central nervous system (CNS).

### 3.1. Application of Molecular Optical Biosensors for Peripheral Processing of Taste

In contrast to artificial taste sensors, devices or recombinant cells, native taste cells detect tastants with a much higher performance and in a physiological context [129]. In search of taste transduction molecules, primary cells isolated from taste buds or lingual slices of animals are thus an indispensable tool. Due to limited accessibility of human samples, these approaches mainly used rodent tissue. However, as taste transduction displays differences between species, our knowledge about human taste physiology, even at the level of primary signal transduction, is still incomplete. Studies on taste signaling were mainly achieved using either dissociated primary taste cells or lingual slices containing taste buds. In both cases, these samples are derived from an acute isolation procedure and could be used only for a few hours. As a readout for sweet, bitter, umami, salty and sour responses mostly fluorescent chemical Ca^2+^ sensors were used. For sour and salty taste, also pH fluorescent indicators were used (Figure 3, Table 1). 

#### 3.1.1. Chemical Ca^2+^ Sensors 

Fluorescent biosensors of divalent metal ions were first discovered by Tsien in the 1980s [130,131,132]. They are commercially available and allow live cell imaging studies upon easy loading protocols. In the field of taste research, Ca^2+^ sensor dyes are preferentially used among ion indicators and have promoted the discovery of main signaling mechanisms (Table 1). A pioneering study used the Ca^2+^ indicator Fura-2 in rat taste cells to prove that bitter substances induce Ca^2+^ release from internal stores [133], and this was confirmed in mudpuppy [134]. Fura-2 is a dual excitation single emission Ca^2+^ indicator whose excitation spectra shifts from a wavelength of ~380 to ~340 nm upon Ca^2+^ binding. Therefore, the ratio of the emissions at those two wavelengths is directly related to the amount of intracellular Ca^2+^ [135], offering the advantage of an internal calibration and relative robustness against uneven dye loading or sample thickness, variable wavelength and intensity of excitation light, dye leakage and photobleaching [130]. With a Ca^2+^ dissociation constant (K_d_) of 140–230 nM, Fura-2 is ideally suited to detect physiological cytosolic Ca^2+^ concentrations [130]. Fura-2 requires ~10–40 min to permeate into the cells. To reduce accumulation in organelles, it is often used in combination with sulfinpyrazone, a blocker of the organic anion transport system [136]. To further improve loading and retention inside the cells, the derivative Fura-2-acetoxymethyl ester (Fura-2-AM) was developed [137]. The non-polar ester permeates the plasma membrane, and the ester bonds are then cleaved by cytosolic esterases, leading to a trapping of the polar dye inside the cell [131]. To inhibit extrusion from cells via multidrug resistance transporters (MDR1), as observed for Fura-2-AM in rat vallate papilla cells, substrates and/or inhibitors of MDR1, such as verapamil, tamoxifen and cyclosporin A can be used [138]. Efficient Fura-2 loading protocols of both, dissociated cells and taste buds, were developed [37,139,140,141,142,143,144]. In combination with conventional and confocal microscopic live imaging, these were used to unravel the Ca^2+^ responses of sweet, umami, bitter, salty, sour, fatty acid and kokumi taste in mice (Table 1). Furthermore, these experiments revealed mechanisms important for Ca^2+^ homeostasis in taste cells and for unconventional Ca^2+^ entry. For example, store-operated Ca^2+^ channels (SOC) were found to be crucial for fatty acid and bitter-mediated signaling [97]. Mitochondria and sodium-calcium exchangers (NCXs) were shown to control basal cytosolic Ca^2+^ levels [145]. In addition, Fura-2-based experiments helped to solve the debate whether VGCCs are involved in Ca^2+^ entry in type II cells. Generally, it was believed that only type III cells possess VGCCs [113,146]. However, with the use of Fura-2, some type II cells were described to utilize a different pathway than PLCβ2/IP3 in response to bitter compounds and to display voltage-gated Ca^2+^ entry [103]. Fura-2-based experiments revealed the modulation of taste signaling by neurotransmitters and hormones, such as ATP [140,144], serotonin [38], glutamate [143], adrenaline [147] and oxytocin [142]. Application of Fura-2 on taste cells from different species, including mice, rats [133], mudpuppy [148] and humans [98,125] unraveled species-dependent similarities and differences. For example, compared to mammalian taste cells, mudpuppy counterparts showed that prolonged bitter stimulation results not only in Ca^2+^ release from stores but also in Ca^2+^ influx [148]. Fura-2 was also used in live imaging experiments in cells derived from a taste 3D culture [127]. 

Thanks to its short wavelength spectrum, Fura-2 can be combined with green fluorescent proteins (GFP) and yellow fluorescent proteins (YFP) to label specific cell populations (see Section 6) [38,42,113,141,143,149,150]. This has facilitated the discovery of functional differences between receptor and presynaptic cells in terms of Ca^2+^ signaling: as mentioned above, for type II cells the major Ca^2+^ source is the endoplasmic reticulum, while in type III cells Ca^2+^ enters via voltage-gated Ca^2+^ channels. Further, Fura-2 was used in combination with the Na^+^-sensitive dye AsanteNaTrump-2 to simultaneously measure Ca^2+^ and Na^+^ changes in response to the activation of TRMP5 and TRPM4 channels by sweet and bitter compounds [151]. The interplay between intracellular acidification and Ca^2+^ signals was investigated by live cell imaging experiments measuring at the same time H^+^ with the pH-sensitive dye pHrodo-Red and Ca^2+^ with Fura-2 [69]. Fura-2 can be further combined with the ratiometric voltage-sensitive dye di-8-ANEPPS although this is not optimal (see Section 4.1) [152]. Fura is also available in a red version (Fura-red) which is excited at 605–700 nm, and when combined with the Ca^2+^ dye Fluo-4 can be used for ratiometric measurements, in order to correct for changes in focal plane which may occur during imaging [153,154].

Besides ratiometric Ca^2+^ dyes such as Fura-2, also single wavelength indicators have been used. They are very bright and can be easily combined with other fluorophores without spectral overlap [155]. These dyes do not shift their excitation or emission spectra upon Ca^2+^ fluctuations, but change in fluorescence intensity according to the intracellular Ca^2+^ concentration. Thus, single wavelength indicators require excitation with only one wavelength and the time intervals between two subsequent image acquisitions is reduced. This is of importance for live cell imaging experiments, especially when measuring fast Ca^2+^ changes such as voltage-gated Ca^2+^ currents generated by action potentials. A disadvantage of single wavelength indicators is that dye loading may differ between cells, compromising a correct Ca^2+^ quantification. Thus, these cells cannot be directly compared in terms of Ca^2+^ levels and each individual cell has to be normalized to its background fluorescence [135]. Amongst the single wavelength indicators, fluo-dyes belong to the family of high affinity Ca^2+^ indicators with a K_d_ of 390 nM [155]. In Ca^2+^-uncaging experiments with Np-EGTA-AM, Fluo-4 served to clarify the presence of Cl^−^ conductance activated by Ca^2+^ in taste cells [156]. In addition, fluo-dyes have been largely used in recombinant HEK cells where Ca^2+^ transients in response to taste stimulation were predominantly analyzed with plate reader-based bulk measurements (see Table 1 and Table 3, and Section 5).

Due to their high quantum yield and low toxicity, Calcium Green-1 (CaG; X 490 nm, M 530 nm) and Calcium Orange (CaO; X 549 nm, M 576 nm) are preferred single wavelength indicators for live Ca^2+^ imaging of intact taste tissues. The K_d_ of CaG is ~190 nM and upon Ca^2+^ binding its fluorescence emission increases up to ~100-fold [155]. In its dextran-conjugated form (CaGD) it is not membrane permeable but has to be applied invasively with a patch pipette [135]. However, once inside the cells, dextran conjugates are retained for long periods and even tissue slicing and cell dissociation do not affect their intracellular concentration. Thus, specific CaGD and CaOD loading protocols were established in the taste field [33,157,158] which allow to specifically label the apical, differentiated taste cells, while basal progenitor cells remained dark. In this way, individual taste cells, their cell bodies and both apical and basal processes could be readily distinguished. This was ideal for studying taste sensation, since differentiated type II and type III cells are the cells responding to tastants with Ca^2+^ signals. Furthermore, this approach permitted to study taste cells in the intact taste bud with apical and baso-lateral polarity for exposure to stimuli in a physiological way [42]. Since desensitization is an issue in primary taste perception, it was also important that stimuli could be applied at slow or fast time scales, i.e., either as bath perfusion or by pressure injection [42,47,102]. Confocal microscopy for live imaging from individual cells was adopted to study Ca^2+^ responses to umami [157], bitter [74,159], kokumi [100], sour [56,59,160] as well as sweet and salty compounds [161]. Furthermore, to enable in situ two-photon live imaging, CaGD-soaked paper was placed on the tongues of anesthetized mice and uniform electrical pulses were applied with a tweezer type electrode to the tongue surface. Intravital two-photon microscopy was then performed on the tongue firmly positioned in a custom-made chamber. This allowed to visualize taste responses upon application of small molecules on the tongue as well as upon their entry into blood circulation, suggesting that small molecules can diffuse through tight junctions to reach microvilli from below to stimulate taste receptor cells in an indirect manner and not only through the taste bud pore [77]. Although CaOD has been reported to be more difficult to load than CaGD and to be poor in performance [59,162], it was used when the green channel was employed for other markers such as in transgenic mice expressing GFP under specific promoters [33,42,49,59,114]. Additionally, CaO was used in combination with the pH-sensitive dye BCECF-D to simultaneously measure pH and Ca^2+^ changes in intact lingual preparations upon focal stimulation [59,163].

#### 3.1.2. Genetically Encoded Ca^2+^ Indicators (GECIs)

A main disadvantage of synthetic Ca^2+^ sensors is their lack of control over loading into specific cell types and subcellular localization. Thus, synthetic dyes sometimes do not enter the cells of interest or they tent to compartmentalize in the endoplasmic reticulum or other compartments [173]. In contrast, genetically encoded Ca^2+^ indicators (GECIs) allow cell-type and subcellular targeting specificity [174,175]. Most GECIs do not contain synthetic compounds or cofactors and are not extruded by the cells. They are mostly fluorescent proteins that change their emission/excitation spectra upon Ca^2+^ binding inside the cells. Synthesis and expression of GECIs occurs upon introduction of genetic expression elements into target cells [174]. GECIs have been engineered after the discovery of aequorin and GFP, both derived from the jellyfish, *Aequorea victoria* [175,176,177,178]. Currently, most genetically encoded biosensors use spectral variants of GFP (XFP) with improved optical properties or siblings of GFP from other cnidarian species [174]. Considering the advantages of GECIs, such as the possibility to target them to a specific cell population and even to subcellular compartments, it is surprising they have never been used to transduce primary taste cells. Maybe, one reason is that the expression of recombinant proteins might require some days and this has to fit into the short life span of taste cells of ~10 days. However, a recent approach using expression of a G-GECO Ca^2+^ sensor in 3D cultures of an immortalized human tongue cell line showed measurements of acute Ca^2+^ changes with confocal and light-sheet fluorescence microscopy upon tastants perfusion [179]. Furthermore, cell-type specific expression of GECIs followed by in situ microscopy was realized in a few studies [55,180,181]. Amongst these, Roebber et al., used *Pirt*-GCaMP3 mice where the Ca^2+^ sensor is expressed in sensory neurons as well as in type II and type III cells, thus this approach is well suited to investigate salt detection in tongue slices [55]. Apart from this, genetically encoded sensors were mostly used to study the decoding of taste sensation in the CNS (see Section 3.2). 

#### 3.1.3. Molecular pH Biosensors 

While Ca^2+^ is the main effector of taste signaling in general, the study of sourness also needs measurements of proton concentrations both in the extracellular milieu and in the cytoplasm (Table 1). Furthermore, the responses to acids are dependent on acid strength, with weak organic acids, such as citric and acetic acids, paradoxically inducing a stronger response [59]. Thus, the ideal setup for analysis of sour taste signal transduction uses contemporary measurements of Ca^2+^ and protons. While Ca^2+^ transients are rapid and Ca^2+^ concentration can increase up to 100 times, the intracellular pH, usually only varies between 6.8 and 7.4 in the cytosol and variations have longer time scales. To quantitatively measure pH, the pKa of the indicator must match the pH of the milieu. Mostly, the dyes pHrodo-Red [66,67], and BCECF [59,163], in their permeant AM form were used as pH indicators in the taste field. BCECF (2′,7′-bis-(2-carboxyethyl)-5-(and-6)-carboxyfluorescein) is a polar derivative of fluorescein introduced by the “three witches” Rink, Tsien and Pozzan in 1982 [182]. The pKa of BCECF of 6.98 makes it suitable to measure intracellular pH changes and its emission fluorescence intensity decreases with acidification. With rare exceptions [59], BCECF is usually applied as a ratiometric dye, as measuring the emission upon excitation at its isosbestic point (440 nm) and the maximum (490 nM) allows to correct for concentration, pathlength, leakage and photobleaching [66,164,183]. In contrast, the fluorescence intensity of the single wavelength indicator pHrodo-Red (pKa of 6.5) increases with acidification and its excitation/emission maxima (560/585 nm) render it suitable for multiplexing with fluorophores such as GFP. Quantification of intracellular pH is possible for both, pHrodo-Red and BCECF, upon calibration with nigerin [164]. In the dextran and AM forms, BCECF is suitable for iontophoretic loading of the lingual epithelium. To simultaneously investigate pH and Ca^2+^ changes in the very same cell in response to sour stimuli, BCECF was applied together with CaOD [59,163]. Confocal microscopy upon focal application of sour stimuli to the taste pore with a puffer pipette allowed to precisely determine the exact concentration of the acid reaching the taste pore region by adding to the stimulus solution the dye lucifer yellow CH, which is pH-insensitive and not cell-permeant [59,163]. It was determined, that the acids reaching the taste pore were able to acidify most of the taste bud cells, but only a subset of them (25 %) showed concentration-dependent Ca^2+^ responses [59,163] at concentrations similar to the sensitivity for citric acid and HCl in mice [184]. This also demonstrated that sour-induced Ca^2+^ transients occur only in type III cells and are mediated by Ca^2+^ entry via voltage-gated Ca^2+^ channels. Furthermore, a linear relationship between intracellular acidification and Ca^2+^ response, but not with the extracellular pH, was shown. At equal acidity, weak acids (undissociated: citric, acetic acid, with higher pK_a_) were a much more potent stimuli than strong acids (dissociated: HCl, with lower pK_a_) [59,163]. As mentioned earlier, the non-linear relationship between extracellular and intracellular pH may be explained by separate pathways for weak and strong acids, with the second involving the entry of protons through the Otop1 channel [69]. Recently, this hypothesis was confirmed with simultaneous live imaging of pHrodo-Red, Fura-2 and YFP in *PDK2L1*-YFP reporter mice [66,67]. This revealed both intracellular acidification by acetic acid and Otop1-mediated proton influx by HCl, the latter with a lower onset and offset compared to acid diffusion. Similarly, pHrodo-Red proved that Otop1 channels mediate intracellular acidification upon lowering the extracellular pH in a recombinant cell model [185]. 

### 3.2. Biosensors to Study Taste Representation in the Central Nervous System

While primary taste signal transduction occurs in taste buds, their responses are transferred to the brain, where they are processed and converted into an internal representation of taste (Figure 1). In the brain the information is processed and passed by several relaying structures, to terminate in the cortex where all the information about palatability (taste quality), reward (hedonic value) and energy source (metabolic information) are integrated to take a decision about food ingestion. Other sensations, including chemesthesis as well as sight and hearing play also an important role in the interpretation of food value and influence our behavior. 

Sensory neurons innervating taste buds have their cell body in the geniculate, petrosal and nodose ganglia and synapse with neurons of the nucleus of solitary tract (NTS) in the brain stem (Figure 1) [186]. These project to the parvicellular portion of the ventroposteromedial nucleus of the thalamus [187]. From there, the information travels to the primary gustatory cortex in a region called “insula”, where the neurons respond to taste stimuli as well as to visual, auditory and tactile cues to decode the qualities of food. In humans, the information passes then to the orbitofrontal cortex, which contributes to perception of tastants, pleasantness of food and decision making about food ingestion [188,189]. In the field of taste neuroscience, a principal question is how signals are decoded at the different levels. Currently, mainly three hypotheses are being discussed. First, the “labelled-line coding” theory proposes that taste cells and neurons are tuned to respond only to a single taste modality and use dedicated neural channels. Second, the “combinatorial-coding” theory suggests that stimuli are decoded by the contemporary activity of a particular ensemble of neurons. Third, each neuron may respond to diverse flavor categories using a “temporal-coding” of action potential firing (reviewed in [81,190]). It is believed that from the periphery to the central nervous system (CNS), there is an increasing complexity of taste code processing, suggesting a preference for labelled-line coding at the level of taste buds and sensory fibers and for combinatorial and temporal-coding involving functional neuronal ensembles in the higher structures (for review [20,191]). While topographic mapping of specific taste qualities to dedicated brain regions was neglected [192], this view was challenged by experiments in mice showing that neurons coding for one taste quality are anatomically clustered [193] and by a magnetic resonance study in humans, which demonstrated that the different taste categories are represented in discrete regions of the insula [194,195]. All these theories were developed based on electrophysiological or optical recordings in living animals (reviewed in [187,196,197]).

A lack in spatial or temporal resolution inherent to the aforementioned approaches was overcome with the use of in vivo microscopy of genetically encoded biosensors as these combine high spatial and temporal resolution. This technique allowed the generation of taste stimulus profiles by localizing individual neurons and quantifying their Ca^2+^ changes, which correlate with the firing of action potentials (Table 2) (reviewed in [198]). The two major technical challenges of such an approach are: (1) genetic targeting of a specific neuronal population, and (2) real-time imaging in brain regions in living animals (Table 2). To address the first point, targeting of genetically encoded biosensors to defined cell populations can be done by stereotactic injection of viral biosensor carriers. Amongst these, Adeno-associated viruses are attractive to induce long-term stable transgene expression in the cells of the CNS with high efficiency and without side effects [199]. They mediate cellular specificity via serotype [200]. Alternatively, transgenic mice using cell-type specific promoters can be generated to express a GECI in the target neuronal population (Figure 3). But a major challenge here is the identification of specific promoters. For example, in one study, 40 transgenic mouse lines were made and screened to efficiently and selectively drive GCaMP3 expression by the promoter *Thy1* [201]. The alternative strategy was to drive GCaMP3 expression in the soma of geniculate ganglia neurons by stereotactic injection of a viral construct in the brainstem. To perform live Ca^2+^ imaging with good spatial and temporal resolution in the ganglia, which are buried in a bony structure, a micro-endoscope was positioned directly into the tissue [201]. As for the second challenge, next-generation Ca^2+^ sensors, such as GCaMP6, detected single action potentials in vivo with high reliability [202]. Upon opening of the skull via surgery to generate an optical imaging window, Ca^2+^ changes were measured in vivo mainly via two-photon microscopy. The latter features low phototoxicity and reduced light scattering and thus permits imaging up to a depth of few millimeters (reviewed in [203,204]) and to record Ca^2+^ changes in real-time at cellular resolution with fields of view of 200–500 µm^2^ [205].

To correctly identify the anatomical structure of interest in the brain, additional tracing experiments are often required. For this purpose, virus particles encoding fluorescent proteins (GFP, DTomato, mCherry or tWGA-DsRed) or dextran-conjugated dyes (MicroRuby dextran) can be injected in structures that send innervation to the selected brain region (anterograde tracing) or receive innervation from them (retrograde tracing) (Table 2) [192,193,206,207]. For example, upon expression in type II cells, the tracer tWGA-DsRed was transferred to neurons via trans-synaptic transfer and revealed that sweet and bitter taste have distinct topographic representation in the NTS [206]. Similar approaches were used to delineate the projections from the gustatory cortex to the amygdala [207], or from the thalamus to the gustatory cortex [192,207]. Another way to trace neurons activated by gustatory stimuli, is to analyze the expression of early genes like *c-fos* or *Zif268* [206,208]. However, this cannot be performed in living mice and has so far involved extensive sectioning and computational reconstruction of 3D images. Recently published methods of optical tissue clearing allow to avoid sectioning, since they render the brain transparent to visualize the tissue in toto by direct 3D imaging [209]. Finally, besides purely descriptive analysis of circuitries, novel optogenetic approaches additionally permit to delete functional connections selectively via targeted diphtheria-toxin expression in freely behaving animals. This approach was used to characterize the role of SatB2 neurons of the parabrachial nucleus in gustatory sensation [208]. SatB2 was found to be a selective marker of sweet-sensitive neurons and upon their ablation in transgenic mice, the sweet taste sensation was severely impaired, while the other taste sensations remained intact. Furthermore, using a miniaturized microscope to observe SatB2-positive neurons expressing the GECI GCaMP6s it was possible to visualize the activity of sweet responding neurons in awake animals during licking behavior. This showed that neuronal activity was synchronized with licking. The expression in SatB2-positive neurons of channelrhodopsin, a light-activated Na^+^ channel regularly employed in optogenetic settings, allowed the specific photostimulation of SatB2-positive neurons and induced a licking behavior similar to that of sweet substances, even when water was presented. This suggested that these mice sensed the sweet taste upon optogenetic stimulation of SatB2-positive neurons, even if the taste buds were not involved [208]. Such complex experiments are important to unravel the central mechanisms of gustatory sense (reviewed in [210,211,212]).

## 4. Additional Optical Biosensors in the Taste Field

### 4.1. Voltage-Sensitive Dyes

Although of their tongue-epithelial origin, taste cells possess electrical characteristics comparable to those of neurons [214]. Patch-clamp recordings showed that mammalian taste cells are electrically excitable and can fire action potentials (AP) [215,216] in response to sweet [217], salty [218], sour [63] and bitter compounds [219]. In rodents, voltage-gated Na^+^ and Ca^2+^ currents are involved in depolarization, whereas K^+^ and probably Cl^−^ currents participate in repolarization during an AP. Type II and type III cells differ in the density and kinetics of Na^+^ and K^+^ currents and in the duration of APs [146,220,221,222]. Since different stimuli evoke distinct patterns of AP firing, these are likely to play a role in taste quality coding [223] and the regulation of neurotransmitter release [214]. Furthermore, the firing activity of taste cells seems to be correlated with that of nerve fibers, and to depend on taste modalities [224]. 

Alternative to patch-clamp studies targeting only a single cell, the electrical activity of multiple cells can also be measured with voltage-sensitive dyes, which have been in use since the 1970s [225]. Their fluorescence changes depend on membrane potential specifically at the plasma membrane [226]. At difference with Ca^2+^ indicators, they have a time resolution in the few milliseconds range, they can measure also subthreshold events and can be used in cells lacking depolarization-induced Ca^2+^ changes [226,227,228]. However, voltage dyes are often less bright, requiring stronger illumination which may lead to photobleaching and phototoxicity [226]. Despite their advantages, voltage-sensitive dyes have been rarely used in the field of taste research. Therefore, it is still largely unknown how voltage changes in taste cells are transduced in Ca^2+^ signals, neurotransmitter release and communication with the afferent fibers. Among the voltage-sensitive dyes, analogues of aminonaphthylethenylpyridinium (ANEP) [229,230] have been the most popular. They fluoresce when incorporated in the cell membrane and can be applied as retrograde tracer of neuronal pathways [231]. ANEPs are “fast potentiometric dyes”, where changes in the electric field affect the electronic molecule structure, resulting in a shifted excitation spectrum [232]. ANEPs can therefore be used to measure membrane potential by dual wavelength ratiometric methods [233,234]. They show a linear electrochemical response to membrane potential changes from −100 mV to +100 mV [230,235] with response times of few milliseconds and they are easily incorporated in the cell membrane upon short incubation [160]. During the first 15–30 min after incubation, the dye response is unstable and can strongly fluctuate in both directions [152]. Quantification of ANEP signals are achieved by ratiometric measurements using blue (450 nm) and cyan (505 nm) excitation [233,234]. In combination with Fura-2, ANEP-based measurements showed that glutamate-induced Ca^2+^ transients in taste cells occur via two mechanisms, of which only one involved cell depolarization [152]. Another study used ANEP to investigate depolarizing potentials in acid-sensitive cells in combination with CaGD and BCECF for the measurements of Ca^2+^ and pH, respectively [160]. Since spectral properties precluded simultaneous measurement of all three parameters, they could only conclude that acidification-induced pH decrease, depolarization and Ca^2+^ changes occur in a subset of cells, but the temporal order and the occurrence in the same cells remained elusive.

Changes in the membrane potential of taste bud cells were optically evaluated with tetramethylrhodamine methyl ester (TMRM), another voltage-sensitive dye. This dye is obtained by esterification of tetramethylrhodamine [236]; it is a lipophilic cation that localizes in the cell membrane of the outer mitochondrial membrane, in proportion to its potential [236,237]. In intact tongue epithelium, taste buds were stained by exposing their basolateral membrane to the TMRM solution, using a special chamber. Comparison of patch-clamp and optical voltage recordings revealed that cells in the taste bud are heterogenous in their voltage responses to a tastant mixture, with some cells undergoing depolarization and others hyperpolarization [238]. At the time when these studies were published, the functional differences between type II and type III cells were not clear yet and many components of the signaling cascade were still to be discovered. It is therefore surprising that no further investigation with voltage-sensitive dyes followed. For example, online voltage measurement in taste cells may help to refine the mechanisms of signal transduction and to link changes in membrane potential to the activation of voltage-gated currents. Additionally, since optical recordings offer the advantage to measure voltage changes in a large cell population, voltage sensors could help to understand the dynamics of cell–cell communication in the taste bud. Furthermore, simultaneous imaging of Ca^2+^ in taste bud cells and voltage in the nerve fibers may help to understand how kinetics of Ca^2+^ transients are transduced into defined firing patterns of afferent fibers.

An emerging technology for optical monitoring of voltage dynamics comes with genetically encoded voltage sensors (GEVIs). GEVIs are cell-specific, less invasive, allow long-term recordings of larger cell populations and give additional information about the anatomy of the structures imaged. GEVIs can detect slow subthreshold events as well as repetitive firing of APs, therefore they are suitable to detect neuronal ensembles and synchronized activities. However, GEVIs are still slower compared to the fast voltage-sensitive dyes (ms vs µs). Furthermore, photobleaching and toxicity might be an issue (reviewed in [239,240]), although new GEVIs have improved sensitivity, brightness and kinetics [241]. GEVIs can be used in vitro and in vivo in order to report voltage from dozens of neurons simultaneously [242] and they can be targeted to a specific cell population [243]. Initial versions of GEVIs were fusions of a fluorescent protein and a voltage-sensing domain. Recent designs, such as Ace-mScarlet A, used a bright fluorescent protein with a voltage-sensitive rhodopsin. Thanks to its red-shifted spectrum, Ace-mScarlett can be used together with green emitting GEVIs or GECIs and also with blue-light excitable opsins, allowing flexible and complex experimental configurations suitable to study neuronal circuits [244]. A “chemigenetic” sensor called Voltron was based on Förster resonance energy transfer (FRET) between a rhodopsin and a synthetic dye. Its increased photon yield allows longer in vivo recordings and in ten times more neurons [241]. Despite the large progress in GEVIs synthesis and their increasing applications, they have not yet been applied to study taste transduction in taste buds or in the brain and only few studies report the use of voltage-sensitive indicators to study functional projections between the gustatory cortex and other brain regions [245]. However, further use of GEVIs is largely desired to improve our knowledge about cell–cell communication within the taste bud and coding of taste signals in the brain. 

### 4.2. cAMP Sensors

Besides Ca^2+^ indicators, also cAMP sensors may be of interest for taste studies as the cAMP/PKA pathway was shown to be involved in bitter, sweet and umami transduction (Figure 2) [43]. Intriguingly, since the discovery of the sweet taste receptor and the PLCβ2/IP3 pathway, cAMP research was largely neglected in taste research. However, in early studies, it was shown that caloric sugars lead to GTP-dependent elevation of cAMP levels. This in turn was proposed to induce PKA-mediated phosphorylation of K^+^ channels resulting in their closure and cell depolarization [88,89,90,91,92]. This view was later challenged by experiments on gustducin knockout mice, which showed that bitter and umami compounds activate PDEs to keep cAMP levels and PKA activity low in order to allow a permissive condition for the taste-induced Ca^2+^ signaling (Figure 2) [43,246,247]. Alternatively, it was proposed that cAMP is important to regulate the activity of cyclic nucleotide-gated channels that play a role in regulating membrane potential [248,249]. Since in these studies cAMP was measured with enzyme immunoassay on lysate extract from tongue epithelia, they provided poor temporal and no spatial or cell-type specific information. However, now there are fluorescent molecular cAMP biosensors that allow to study cAMP dynamics in living cells. The first molecular cAMP biosensor was introduced in 1991 [250]. It used fluorescein labelled catalytic- and rhodamine labelled regulatory-subunits of PKA to produce a FRET-based cAMP sensor [250]. However, since the fluorophores are not genetically encoded, FRET-sensors had to be microinjected [251]. In the last 20 years, genetically encoded FRET-based cAMP biosensors were developed and allowed to unravel the relevance of compartmentation of intracellular cAMP signaling. The first genetically encoded cAMP sensor was generated tagging the PKA with two FRET-suited GFP mutants and this was used to monitor cAMP fluctuation in cardiac myocytes [252,253]. Since then, cAMP sensors have been continuously improved including newly discovered cAMP binding sequences and fluorescent proteins (for reviewed see [254]). Furthermore, subcellular targeting of such sensors permits to study changes in cAMP levels in micro- and nanodomains [255] and the genetic approach renders them suitable for studies in recombinant cells lines, primary cells and in vivo [256,257,258]. Therefore, they would be very helpful to define the precise role of cAMP in taste transduction. For example, they could help to understand if natural sugars and synthetic sugars differ in the stimulation of the cAMP pathway [91,92,139] and if sweet, bitter and umami signaling induce diverse cAMP responses. 

## 5. Recombinant Systems to Study Taste Receptor Function

Engineered reporter cell assays systems have been developed and used in the field of taste to characterize the functions and structure of TRs and to find novel agonists and antagonists. In particular, the human embryonic kidney cell line 293 (HEK293) [47,72,259,260,261] and less frequently also Chinese hamster ovary (CHO) [262] cells were used for this purpose (Figure 3, Table 3). Several transduction components of the different taste modalities have been so far expressed in heterologous systems, such as sweet, umami and bitter receptors, but also channels including V1R [261], CaSR [263], and PKD2L1 (Table 3) [154,260,264]. About 20 years back, pioneering studies in this field faced two major challenges: first, the understanding of signaling pathways was still very limited, and second, targeting the receptor proteins to the plasma membrane was not efficient. These problems were addressed by co-expression with the α-subunit of the G-protein gustducin [83,94] and the production of chimeras between the taste receptor and a sequence of rhodopsin [47]. These studies were crucial to understand that T1R2-T1R3 dimers form functional sweet taste receptors [68] and that homomeric T2Rs act as bitter receptors [47]. Furthermore, these studies proved that both sweet [83,94] and bitter [47] receptors require the α-subunit of gustducin to give rise to intracellular Ca^2+^ signaling via activation of PLCβ2. Further, recombinant systems allowed to identify the structural requirements for bitter receptor activation and agonist interaction [265] and they were used to explain the mechanism of bitter side-taste of synthetic sweeteners: indeed, while acesulfame and saccharin at low concentration are sweet agonists, they become sweet antagonists and activate bitter receptors at higher concentrations [72]. Recombinant systems were also used to study the interaction site of the sweet receptor with the sweet taste inhibitor lactisol [120] or with synthetic sweeteners such as cyclamate [266]. They were also crucial to prove that sweet and umami receptors share the T1R3 subunit [93]. Heterologous systems expressing the transient receptor potential family members PKD1L3 and PKD2L1 were used to investigate whether they may represent the sour taste receptor [150], and recombinant systems were important to prove that TRPM5 opening is induced by intracellular Ca^2+^ [112,267]. While the most intensely studied taste quality in engineered cell lines is bitterness, reports for salty taste using recombinant systems with optical readouts are lacking. 

Since the discovery of the composition of taste receptors, they have been co-expressed with the Gα15 or the artificial chimeric Gα16 subunits of gustducin 44, which efficiently couples G_s_, G_q_, G_i_ and gustducin to PLCβ2 [47]. However, since these subunits were derived from murine hematopoietic cells [268,269], the human receptors expressed in the recombinant system did not couple with their native G-protein [270,271]. Furthermore, in native tissue, taste signaling involves also G_αi-2_, G_αi-3_, G_αs_ and G_α14_ which are often co-expressed with gustducin [272,273,274,275], and recombinant assays may suffer from molecular crowding resulting in false-positive results [19]. These may be reasons why the results obtained in recombinant systems fall short of the situation in vivo, especially for human cells. Nevertheless, the Ca^2+^ transients induced by the coupling of the recombinant TRs to promiscuous G-proteins can be quantified with synthetic Ca^2+^ indicators, of which Fura and Fluo are the most preferred for their easy loading protocol and high signal-to-noise-ratio (Table 3). However, as GFP was often used to label transduced cells [47,83,94,259,276], Ca^2+^ measurements were contaminated by GFP fluorescence [277]. An approach to overcome this problem is to measure Ca^2+^ changes using red-shifted single wavelength Ca^2+^ indicators like X-Rhod (X 580 nm, M 602 nm) [262,277]. Alternatively, the labelling of transduced cells could employ other fluorescent protein variants [112]. As recombinant cell lines can be easily expanded and passaged, Fluorescent Imaging Plate Reader (FLIPR) approaches to quantify Ca^2+^ changes for enhanced throughput screening were developed to perform quantitative optical screening and cell-based kinetic assays [278]. They allow simultaneous fluorescence measurements from entire 96, 384 or 1536 well microplates, with kinetics in the sub-seconds range in combination with precise compound injection and controlled temperature and CO_2_. However, since FLIPR measurements are at the expense of spatial resolution, additional microscopic imaging is required to get cellular and subcellular resolution (Figure 3) [154,260,264]. 

Industrial applications in the taste field have largely profited from recombinant models, and different cell-based assays were patented to test bitter, sweet and umami substances (reviewed in [19]). For example, HEK293 cells expressing T1R2-T1R3 receptors where used to discover novel sweet taste enhancers or natural high-intensity sweeteners. A bitter-sensitive reporter cell line expressing T2R44 was used to identify the first synthetic bitter blocker GIV3727 [279]. In addition, recombinant cell lines to find novel ligands for umami (T1R1-T1R3) [93] and kokumi receptors (CaSR) [263] were developed. To study salty taste compounds, HEK293 cells transiently transfected with ENaC were used in combination with ion-sensitive dyes as a measure for membrane potential or sodium [280,281]. Given that the functionality of signaling molecules is sensitive to the cellular environment, there has always been a desire in the taste field to work with primary cells. Indeed, human primary taste cell cultures were developed [98,125] and albeit their limited accessibility and life span, the introduction of foreign genes is possible and new techniques are still under development [282]. For example, *α1* adrenergic receptors were expressed in isolated primary taste cells and then stimulated with noradrenaline upon loading with Fura-PE3, which was preferred as it is less prone to leakage and compartmentation [283]. To generate large numbers of human cells for enhanced throughput analysis, an original approach was the immortalization of primary human lingual cells to produce cell lines that endogenously express taste receptors. These were found to express functional bitter receptors as well as naturally occurring downstream signaling components [117]. Additionally, these cells were engineered to stably express a GECI (G-GECO). This prompted the generation of spheroids where Ca^2+^ signals could be measured in all the regions of a spherical 3D structure, where cell–cell interactions are well preserved [179]. Spheroids can be easily and robustly produced by the “liquid overlay” or “hanging drop” technique in specific microplates that can be used for plate reader [284,285] or microscopic measurements [179,286]. Alternatively, the use of non-taste cell lines expressing endogenous taste receptors and downstream signaling molecule can be of option. Since TRs have been found in multiple other tissues as the lungs, gastrointestinal tract and epithelium [287], cell lines as 16-HBE, HGT-1 or HuTu-80, which express natively components of taste signaling, have been used in patens from diverse companies (for overview see [19]). A limitation of this approach is that these cell lines often co-express sweet and bitter receptors, at difference with taste receptor cells.

## 6. Reporter Genes to Mark Specific Taste Cell Populations in Mice 

Reporter genes are readily detectable gene products like β-Gal, luciferase or fluorescent proteins attached to a promoter sequence of another gene whose expression pattern is known or of interest. When used in transgenic organisms, the reporter gene product should thus allow to follow the activity pattern of the employed promoter [47,83,94,259,276,295]. Therefore, reporter genes can help to identify defined cell populations, if the used promoter sequence is specific for these cells and it allows to perform cell-type specific functional studies [296]. The first gustducin reporter mouse used the expression of the *lacZ* gene [111], but before these were introduced to study taste-specific questions, the only way to link a functional response to a defined cell type was to fix the investigated tissue/cell after live imaging and then proceed with phenotypic characterization, e.g., by immunostaining; the same cell had to be found and related to the imaging data. For example, Caicedo et al. aimed to study Ca^2+^ signaling in gustducin-positive cells. To do so, live imaging experiments were conducted, and afterwards the analyzed cells had to be re-identified by anti-gustducin immunostaining [159]. Thus, reporter gene approaches clearly facilitate the identification of the right cell population in live tissue. Currently, GFP is the most widely used reporter gene in the field of taste research (Table 4). The first GFP-based taste-related reporter mouse was presented by Huang and colleagues who exchanged the *lacZ* gene with a red-shifted GFP (BRL) from Wong’s cassette [79]. This achieved GFP expression in 95% of *gustducin*-GFP-positive cells [79]. Since the expression of the reporter gene product depends exclusively on the cell-type specific promoter, the recombinant promoter still drives the expression of the reporter protein in a specific cell upon knockout of the native gene product. Following such a strategy, it was possible to investigate taste signaling in defined cell types in absence or presence of key molecules in response to the different taste modalities (Table 4). Bitter responsiveness in the absence of gustducin was investigated by creating a reporter mouse in which GFP was expressed under the gustducin promoter, but with gustducin itself being knockedout [141]. Another reporter gene promoter to mark type II receptor cells was *PLCβ2*, which was used to create a knockin mouse expressing *PLCβ2*-GFP. Upon loading with CaOD, this allowed to demonstrate responsiveness GFP-positive cells to bitter compounds [297]. *PLCβ2*-GFP reporter mice were also used along with *GAD*-GFP mice. GAD is an isoform of the glutamic acid decarboxylase which is used as a marker for type III presynaptic cells [298]. Measuring Ca^2+^ responses in *PLCβ2*-GFP and *GAD*-GFP cells with CaOD significantly improved the understanding of the differences between type II and III cells in terms of expression of voltage-gated channels [49], signal transmission [114], neurotransmitter stimulation [33,42] and oxytocin-mediated modulation [142]. Further, *IP3R3*-GFP and *TRMP5*-GFP reporter mice were used to identify type II cells [50,113,299], and experiments with Fura-2 showed that only GFP-negative cells responded to depolarizing stimuli, thus confirming that type II cells do not possess voltage-gated Ca^2+^ channels [113]. To discriminate between sweet and umami taste cells (which are both of type II), mice expressing a *T1R3*-GFP fusion protein were generated (Table 4) [34,113,300,301,302,303,304,305]. As bitter is transduced by multiple receptors, the copy number of any individual bitter receptors might be rather low and therefore single T2R constructs may not be feasible. However, Oka and colleagues generated reporter mice expressing the blue-shifted GFP-derivate Sapphire under the control of the *T2R32* promotor [46,306]. Sapphire has improved folding properties, is more pH stable and has excitation/emission peaks at 399/511 nm [307], thus allowing the combination with green-emitting Ca^2+^ indicators such as CaG. This showed, that type II cells expressing bitter receptors are also responsive to high KCl salt concentrations [46]. To address sourness and saltiness, *PKD2L1*-GFP [67] and *ENaC*-GFP reporter mice [48], respectively, were generated. In most reporter gene assays in the field of taste research, Ca^2+^ was measured with synthetic dyes (Table 4), either Fura-2 [277] or CaOD [33,49,114,297]. Alternatively, electrical recordings were used to assay Ca^2+^ transients [67,301,308,309]. Finally, one study presented mice where the GECI GCaMP3, expressed under the *Pirt* promoter, is present in type II and type III cells to investigate salt transduction [55,310].

## 7. Biosensor Cells to Determine Neurotransmitter Release from Taste Bud Cells 

In general, biosensor cells (BCs) are living cells that detect the presence and concentration of a certain analytical substance. In taste research, they were used to study the response of taste cells to a stimulus, by reporting the release of neurotransmitters (Table 5) [315]. As some cell lines express a large variety of different native receptors, they can be directly used as sensor for certain ligands [316]. If the receptors of interest are not endogenously expressed, the cells can be genetically modified by heterologous expression [317]. In addition to the receptor, a biosensor cell-based system needs a transducer to report receptor activity to the experimenter. Often, BCs take advantage of intracellular signaling cascades activated by the receptor-ligand interaction, which amplifies the signal and facilitates its detection and measurement (Table 5) [316]. Accordingly, Ca^2+^ measurements with synthetic indicators are widely used (Table 5) because of their low cost and high efficiency and sensibility. BCs can be flexibly designed and are cheaper than antibodies or enzymes [318], but issues like inhomogeneous expression of receptors, depressed sensitivity and reproducibility need to be addressed [129]. As outlined below, BCs were useful tools to study the release of neurotransmitters and intercellular communication in taste buds (Table 5).

In a first attempt, BCs for serotonin (5HT) were made. These used CHO cells transiently expressing human high-affinity 5HT_2C_ receptors, which trigger activation of PLCβ2, generation of IP3 and Ca^2+^ release from intracellular stores [319]. Thus, serotonin release could be assessed with these BCs by quantifying Ca^2+^ signals with Fura-2 live cell imaging (Figure 3) [320]. A second generation of serotonin BCs consisting of CHO cells stably expressing 5HT_2C_ receptors were used to prove that 5HT is released by taste bud cells upon stimulation with bitter, sweet, or sour compounds [36]. To asses this, a Fura-2 loaded BC was positioned close to a freshly isolated mouse taste bud. Stimulation with KCl, sweet, sour or bitter compounds activated the BCs, indicating release of 5HT from taste bud cells (Figure 3). Since CHO cells natively express some purinergic receptors, the antagonist mianserin or the desensitization with a preincubation in high concentrations of ATP were used to differentiate serotoninergic from purinergic responses [36,105]. Notably, these BCs sensed 5HT concentrations in the nanomolar range, and thus allowed to detect the release of 5HT even from single dissociated taste cells [36,37]. Of course, the procedure was rather delicate and needed great care to position the BCs in immediate vicinity of the target cells to not lose the response. Furthermore, since only about one third of the trials were successful [36], stimulating preselection procedures of the best BCs prior to use with taste cells were applied [41]. 

In a similar fashion, BCs to sense ATP [37], noradrenaline (NA) [39] and GABA [40,41,104] were generated by transducing CHO cells with the purinergic receptors P2X_2_ and P2X_3_ [37,321], α1-adrenergic receptor (α_1A_) [39,322] and heteromeric GABA_B_ receptors plus G_αqo5_ (Table 5) [40]. Conversely, ATP-sensitive BCs based on COS-1 cells utilized their endogenous expression of P2Y receptors coupled to Ca^2+^ mobilization [32,323]. In this approach, a glass pipette was used to position the taste cell. This is advantageous, since the same pipette can be used for patch-clamp recordings to characterize the cell type and to trigger neurotransmitter release also by electrophysiology [32,323]. In the further course, dual BCs were generated (Table 5) by co-expressing two different receptors such as 5HT_2c_ with α_1A_ [39] or with P2X_2_/P2X_3_ [41,56,324]. Upon desensitization of one pathway, these dual BCs could be used to test alternatively one of the two neurotransmitters [40,41]. In some experiments, Ca^2+^ changes were simultaneously measured with Fura-2 in the taste cell and in the BC and the observed delay between the two transients was interpreted as a confirmation for communication via neurotransmitter [105]. 

BC-based research critically contributed to the understanding of the complex cell–cell communication in the taste bud. They unraveled the secretion of ATP by type II cells via an unconventional mechanism mediated by hemichannels [37], the release of 5HT, noradrenaline and GABA by type III cells upon voltage-gated Ca^2+^ entry [36,39,40,41] and of GABA by type I cells upon substance P stimulation [104]. Furthermore, the origin and the modulatory effects of glutamate [143], calcitonin gene-related peptide [324], adenosine [325], GABA [104] and acetylcholine [42] on taste cell-mediated neurotransmitter release were investigated. To develop BC-based systems for higher throughput, it could be advantageous to create co-cultures of biosensor cells and taste cells. In this case, rather than using a chemical indicator, it would be preferable to express genetically encoded sensors for Ca^2+^ or other second messengers such as cAMP [326,327] in the BCs and taste cells. This would be very useful in case of the generation of 3D cultures where loading with synthetic dyes may not be as efficient and homogenous as in 2D. 

## 8. Conclusions and Future Perspectives

In taste research, two main streams run in parallel and are interconnected: one is the study of the mechanism underlying gustation, the other deals with the generation of artificial taste systems that can sense and discriminate between different tastants. Our concluding discussion will first focus on new potential approaches to discover the still missing pieces of the complex puzzle of taste physiology and to further refine the previous findings adding more details on the intracellular signaling and cell–cell communication. In a second part, ideas for novel sensing systems will be proposed that might mimic more closely the in vivo situation, but would be also compliant in terms of cost, technology and ethics. 

### 8.1. New Approaches to Study Taste Physiology 

The last 20 years have seen massive improvement in understanding taste physiology, both in the periphery and in the CNS as well as at the molecular, cellular and network levels. However, most studies focused on the ubiquitous intracellular signal Ca^2+^, potentially leading to an underestimation of other important factors. Indeed, the taste transduction pathways might be much more complex than the models proposed in most reviews including the pathway shown in Figure 1. For example, many knockout mouse models suggest that there may be additional signaling pathways involved, since taste sensation, nerve response and preferences for test compounds are often not completely abolished by eliminating a critical component of the canonical pathway including taste receptors, gustducin or TRPM5 channels [76,86,300,330]. Looking only at Ca^2+^ may not suffice to discriminate between these different pathways, as they may all induce a similar Ca^2+^ increase but with different underlying mechanisms. Along these lines, G-proteins other than gustducin are likely also involved in the signaling cascade [275,331] as well as other second messengers than Ca^2+^, such as cAMP [141,165]. Moreover, sweet signal transduction mechanisms may exist that bypass the activation of sweet taste receptors and instead use glucose transporters (GLUT) [86,87,332]. The pathway usage might be different even for compounds falling into the same taste modality [139], as it might be true for nutritive sugars and synthetic sweeteners or bitter compounds utilizing a PLCβ2-independent pathway at higher concentrations [333]. In this context, it would be helpful to use genetically encoded molecular biosensors for different signaling components such as cAMP and glucose [252,334] or a plethora of other factors (for a comprehensive database, see https://biosensordb.ucsd.edu). Via standard genetic engineering tools, such biosensors can be expressed in recombinant tongue-derived cell lines [179], introduced into primary taste cells via viral infection or electroporation or in transgenic mice for in situ or in vivo studies.

Next, it is widely recognized that specificity of second messenger-based cellular signal transduction involves local organization, by restricting signals to specific organelles and even to microdomains [335]. Until now, this aspect of signal compartmentation has been completely neglected in the investigation of taste transduction, although it is well known that taste cells use many different sources of Ca^2+^ [49,113]. Accordingly, although some taste responses are mediated by Ca^2+^ influx into receptor cells [50,117], the underlying molecular mechanism is unknown, since these cells do not express voltage-gated calcium channels. Finally, measurements of cytoplasmatic Ca^2+^ in combination with drugs that empty the endoplasmic reticulum or block the activity of mitochondria showed an important role of these organelles in the Ca^2+^ homeostasis within the taste bud [168,169]. As valuable as these insights were, future approaches should measure Ca^2+^ signals in the corresponding compartments by appropriate subcellular targeting of genetically encoded probes [336]. In synthesis, switching from chemical dyes to genetically encoded biosensors and adding other signaling components to Ca^2+^ will help to further improve our understanding of the taste signaling cascades.

### 8.2. Development of New In Vitro Taste Systems

Taste cell-based biosensors have a great potential for either basic research or for industrial applications, such as food safety and discovery of new ligands. On the one side, biosensor systems highly specialized for one taste modality are good testing platforms for discovering new agonists and antagonists to specific receptors/ions. Embedded into analytic devices they might serve to detect for example toxins [129] or to screen compound libraries [271,279]. On the other side, taste cell-based biosensors able to respond to flavors of different modalities could interpret complex flavor patterns and may be useful for quality control of food, drugs and the development of new palatable mixtures, such as in search for new sweet beverages with low caloric content. 

On the long run, a complex taste cell-based artificial tongue system can be envisaged, which should fulfil the following criteria. It should 1) reproduce the complexity of the taste bud with respect to structure and functionality, 2) be representative for human gustation and 3) be implementable for high throughput. Considering the complexity of taste buds, 3D taste cell cultures with heterogeneous cell populations might come closest to such an idea. For reasons of reproducibility, these 3D culture could be formed as spheroids [179], cell chip cultures (Figure 4), and organoids derived from progenitor cells [127] or microfluidic devices [337]. Spheroids, prepared with tongue-derived human cells (HTC-8) by liquid overlay technique on commercially available low-attachment plates, were successfully used to measure real-time responses upon application of bitter compounds [179]. Comparable results were obtained upon 3D cultivation of HTC-8-G-GECO cells in special Dynarray cell chips, which can be used for live imaging experiments in perfusion conditions, as they are made of transparent materials and have porous walls (Figure 4, original data from us). Furthermore, the chips can be used in bioreactors with circulating media and with the option of different apical and basal perfusions to favor cell polarization. Further, organoids were generated from Lgr5-positive mouse progenitor cells, which could be expanded and differentiated into the different taste cell types [126,127]. However, neither the purification of such multipotent cells from human tongue, nor the generation of mature taste cells by differentiating human induced pluripotent stem cells have been explored so far. Recent progress in the mechanisms of taste cell development [124,127,128] suggest further steps in that direction. Another effort to recreate a multicellular taste cell-based system with a potential integration in a microfluid device for industrial application, has been recently made with the help of tongue extracellular matrix [337], which was generated from porcine tongue and used to prepare primary 2D and 3D taste cell cultures and co-cultures with neurons [337]. This improved both the expression of gustatory markers and the functionality of taste cells, furthermore taste cell signals were transmitted to the neurons. These findings were applied to test wine [10].

Finally, on the road to an artificial tongue system, the development of optimal cell culture conditions needs to be flanked by novel sensor technologies for more specific and comprehensive readouts. For example, optical and electrical sensors could be used in parallel. Patterns of Ca^2+^ changes in combination with voltage current recordings may allow to discriminate not only between the different taste modalities but also between compounds of the same families, such as natural versus artificial sweeteners. Novel, sophisticated imaging systems suitable to acquire multiple fluorescence signals from microplates with cellular resolution are now available and high-content imaging systems allow the acquisition of confocal images in thick tissue/probes up to a 1536 wells format with multiple lasers and dichroic filters [338,339]. In addition, optical tissue clearing protocols start also to be applied to 3D cultures, which could largely improve expression analysis in three dimensions [340]. For enhanced throughput of electrical recordings, new devices such as multielectrode arrays or automatic patch-clamp platforms allow to clamp hundreds of cells simultaneously [341]. 

In summary, the use of taste cells to develop biosensor systems for gustation is a new field with a wide range of possible applications [5,6,7,9,205]. Next-generation biosensor systems could now be developed with 3D taste cell cultures that can be used for both high-content imaging and live imaging microscopy. The use of genetically encoded biosensors to measure different signaling molecules in defined cellular compartments will allow higher flexibility, easier handling and more specific readouts. A major challenge will be to couple 3D cell cultures expressing such sensors to microdevices able to read optical and electrical signals, in order to develop portable and user friendly “pocket tongues”.

## 9. Literature Research

Literature research used the electronic databases Google Scholar and PubMed, which were screened by two independent researchers for peer-reviewed publications written in English. The selection of articles was carried out by evaluating first heading and abstract, then reading the whole text. Search terms used were “taste”, “gustation”, ”calcium”, ”pH”, ”voltage”, ”dye”, “genetically encoded sensor” and “biosensor”: such terms were differently combined with the Boolean “AND” operation. Most hits were found with “taste AND calcium”. We included all articles without time restriction, while patents were excluded. Further, only studies concerning gustation in the taste bud and in the CNS were included. Conversely, studies in extraoral tissues and in specific brain regions in respect to food reward, expectation and decision making were omitted. Only articles applying optical biosensors in the context of an optical functional readout, such as plate reader or microscope, were included in the table. Other functional read out such as electrophysiology, or studies using only immunohistochemistry or biochemistry were mainly not discussed.

## Figures and Tables

**Figure 1 sensors-20-01811-f001:**
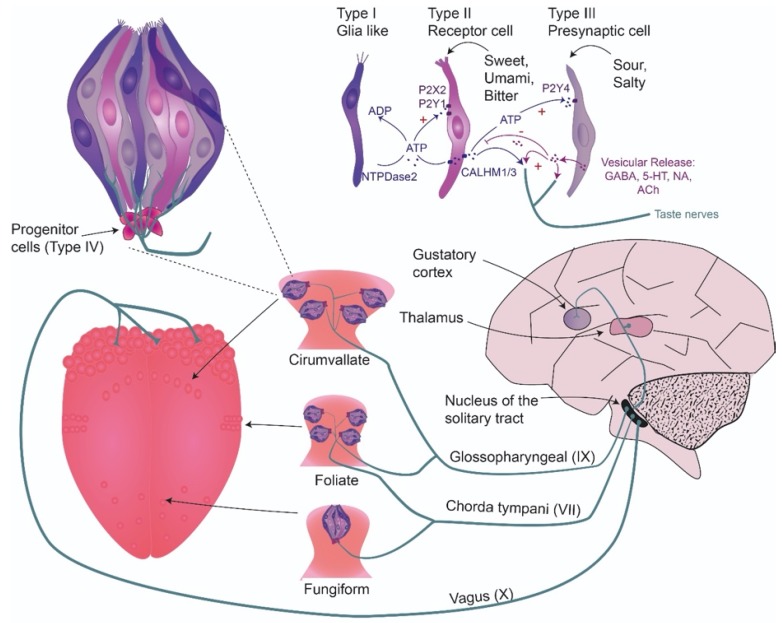
Schematic overview of cells, tissues and organs involved in taste perception. In the periphery, primary taste transduction uses different receptors and intra- and intercellular pathways (indicated) that occur in taste cells of type I–III (upper right). These are organized in taste buds (upper left), which are organized in circumvallate, foliate or fungiform papillae (center) of the tongue (lower left). From there, sensory information is passed on by glossopharyngeal, chorda tympani and vagus nerves (lower right) to the brain (right). Here, signals first arrive at the nucleus of the solitary tract in the brain stem, from where they get relayed to the thalamus and then to the gustatory cortex for final processing.

**Figure 2 sensors-20-01811-f002:**
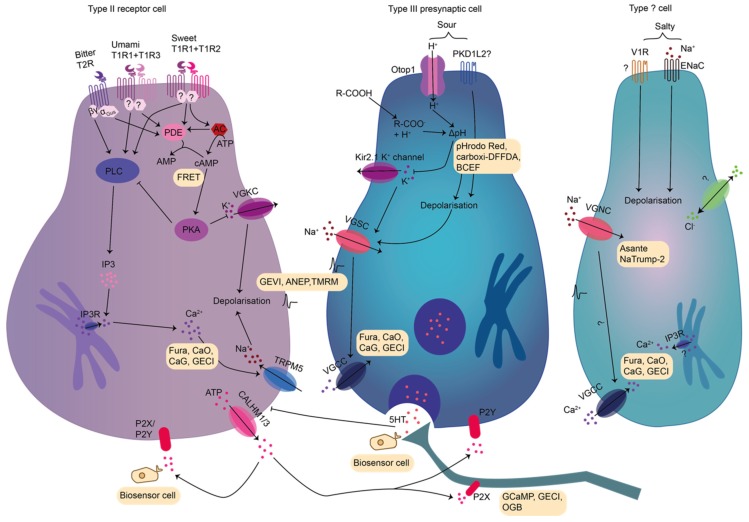
Taste transduction of the five basic taste modalities. Schematic overview of the currently known principal signaling pathways involved in primary taste transduction occurring in taste cells (type as indicated) and their output to neighboring taste cells or gustatory nerve fibers (lower part). The use of molecular fluorescent biosensors for specific research questions, as described in the main text, is indicated here by yellow boxes. For detailed descriptions of signaling pathways and the use of biosensors, please refer to the main text. Elusive aspects are indicated by question marks. Abbreviations of signaling molecules: 5HT: serotonin, AC: adenylate cyclase, CALHM: calcium homeostasis modulator channel, ENaC: epithelial sodium channel, IP3: inositol-3-phophate, IP3R: inositol-3-phosphate receptor, Kir2.1: inward-rectifier potassium ion channel 2.1, P2X/P2Y: purinergic receptors, PDE: phosphodiesterase, PKA: protein kinase A, PKD1L: polycystic-kidney-disease-like ion channel; PLC: phospholipase C, TRPM5: receptor potential M5 channel, V1R: vanilloid receptor 1, VGCC: voltage-gated calcium channel, VGKC: voltage-gated potassium channel, VGSC: voltage-gated sodium channel, α_Gus_: α-subunit of the G-protein gustducin.

**Figure 3 sensors-20-01811-f003:**
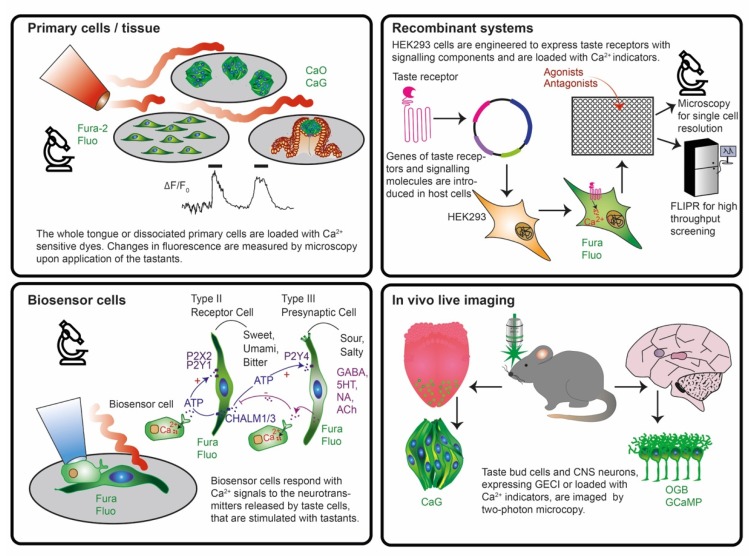
Biological models taking advantage of optical biosensors to study taste signaling. Schematic overview of the four major types of test systems used so far to investigate taste perception. Frequently applied molecular optical biosensors are depicted with green letters, curly red lines indicate local application of tastants. For further details, please refer to the main text.

**Figure 4 sensors-20-01811-f004:**
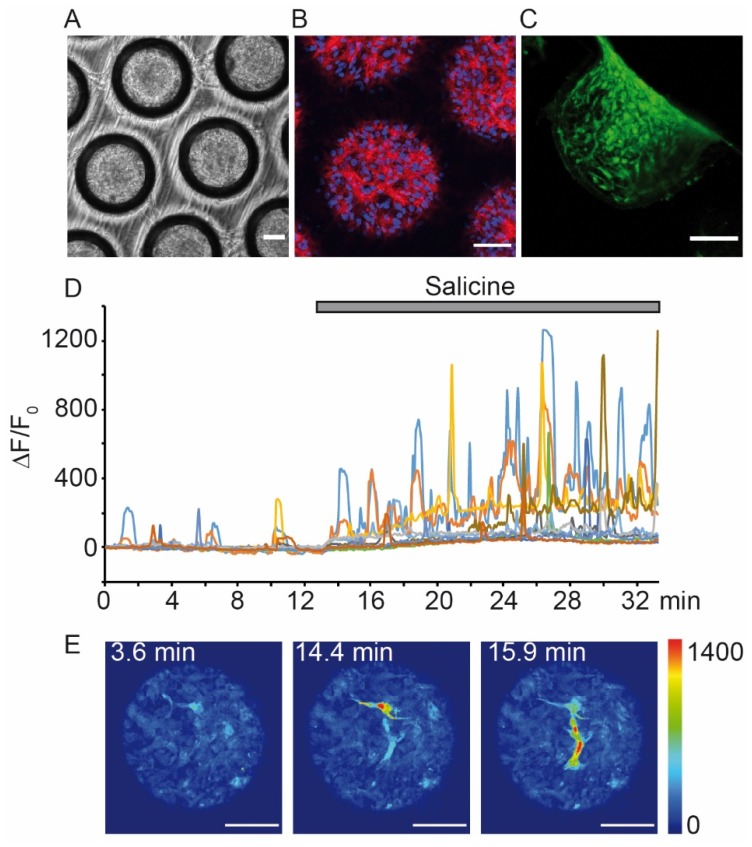
Use of a 3D culture with immortalized human tongue cells expressing a GECI to study Ca^2+^ responses upon perfusion of a bitter compound. Dynarray chips (300 Microns) were filled with 5000 HTC-8-G-GECO cells/cavity and cultured for three (**A**–**C**) or seven days (**D**,**E**). (**A**) Bright-field image. (**B**) Representative confocal fluorescence image of one optical plane at 30 µm from the top of the chip. Upon fixation, cells were labelled with markers for f-actin (phalloidin, red) and nuclei (DAPI, blue). (**C**) Confocal micrograph showing a side view of a chip cavity. Before microscopy, the chip was PFA-fixed and sliced. (**D**) Representative time course of HTC-8-G-GECO fluorescence changes (ΔF/F_0_, normalized to the time before compound perfusion). The chip was mounted in an Ibidi perfusion slide, stabilized with gelatine pads (Scaffolene, from Freudenberg) and then perfused with salicine (20 mM) (all procedures analogous to [179]). Traces show G-GECO fluorescence changes as a function of time, the time period of salicine perfusion is indicated by a grey bar. (**E**) Examples of pseudo-colored confocal images during perfusion in control (left panel) and in salicine solutions (middle and right panels). Blue and yellow/red cues indicate low and high G-GECO fluorescence intensities, respectively. Time points of image acquisition correspond to graph in (**D**). Images were acquired with an inverted Leica TCS SP8 confocal microscope. Scale bars in (**A–C**) and (**E**), 100 µm. (Unpublished data E.M. and T.C.)

**Table 1 sensors-20-01811-t001:** Fluorescent metal ion dyes used in taste research. Different indicator dyes have been used to study taste signaling in tissue/cell preparations. The readout was based on different microscopy techniques.

Cell/Tissue	Species	Imaging	Sensor	Detect	Stimuli	Microscopy Technique	Source
Dissociated taste cells	Hamster	Ex vivo	BCECF-AM	pH	Sour	Conventional	[164]
Single fungiform papilla	Rat						[66]
Slices	Mouse		BCECF-D/AM + CaO + lucifer yellow	Ca^2+^, pH		Confocal	[59]
Epithelium	Mouse		CaGD	Ca^2+^	Salt		[46,48]
Dissociated cells, slices	Mouse				Sour, kokumi bitter, umami, K^+^		[56,100,114,159]
Slices	Rat				Sweet umami bitter, salt, sour		[74,157,161,165]
Isolated taste buds	Rat		CaG-AM		Sour	Conventional	[166]
Isolated taste buds, slices	Mouse *PLCβ2*-GFP		Fura-2 CaGD/CaOD		Adenosine		[42]
Dissociated cells, isolated taste buds, slices	Mouse				GABA	Confocal	[41]
Dissociated cells	Mouse *GAD*-GFP		Fura-2		Sour	Not stated	[150]
Dissociated cells, isolated taste buds	Mouse				ATP	Conventional	[140,144,167]
Dissociated cells, cell aggregates	Rat				Bitter		[50,103,133,141]
Dissociated cells	Mouse				Bitter, umami, sweet, ryanodine	Confocal	[37,168,169,170]
Dissociated cells, isolated taste buds	Mouse *PLCβ2*-GFP				Glutamate		[143]
Dissociated cells	Mouse GAD67-GFP				IBMX, Forskolin	Conventional	[149]
Dissociated cells	Mouse *T1R3*-GFP/*TRPM5*-GFP				K^+^		[113]
Dissociated cells	Mouse				Bitter, adrenergic agonist, K^+^		[147]
Dissociated cells, isolated taste buds	Mouse				Oxytocine	Not stated	[142]
Dissociated cells, isolated taste buds	Rat				Sweet, Forskolin	Conventional	[139]
Dissociated cells	Mouse/Mudpuppy				Bitter		[171]
Dissociated cells	Mouse/Human				Fatty acid		[98]
FACS isolated CD36 pos. cells	Mouse	Primary culture	Fura-2-AM		Fatty acid,	Confocal	[97]
Primary culture of taste cells	Human				Sweet, bitter	Plate reader	[125]
Dissociated cells	Mouse	Ex vivo	Fura-2 AsanteNaTrump-2	Ca^2+^, Na^+^	Bitter, sweet, umami	Confocal	[151]
Slices	Mouse		GCaMP3 in type II and III cells	Ca^2+^	Salt + AF-568,647 or fluorescein		[55]
Tongue	Mouse	In vivo	CaGD		Sweet, salt, sour, bitter	Two-photon	[77]
3D culture (organoids)	Mouse	2D cell culture	Fura-2		Sweet, salt, sour, bitter	Conventional	[127]
Isolated taste bud	Chicken	Ex vivo	Fluo-4-AM		Bitter, umami	Confocal	[172]
Dissociated cells, isolated taste buds	Mouse		Fluo-4M Np-EGTA-AM	Ca^2+^+ uncaging	ATP	Conventional	[156]

**Table 2 sensors-20-01811-t002:** Biosensors used to study taste in the brain. Taste bud are innervated by sensory neurons that convey the information to the CNS. This has been studied with live imaging microscopy in vivo with mostly genetically encoded Ca^2+^ sensors. Abbreviations: NTS: solitary tract, PBN: parabranchial nucleus, GC: gustatory cortex, genic.gangl: geniculate ganglion, Tr.: transgenic.

Region	Species	Transgenic Model	Tracing	Sensor	Detect	Stimuli	Microscopy Technique	Source
Genic.gangl.	Mouse	Tr. mouse *5HT_3A_*-GFP		Fura-2/Fluoro-Gold	Ca^2+^	5HT	Confocal	[38]
		Tr. mouse *Pirt*-GCamMP3		GCaMP3		Sweet, bitter, umami, salt, sour		[205]
		Tr. mouse *Thy1*-GCaMP3	AVV-GCaMP3 (retrograde/NTS)				Two-photon	[201]
NTS		Tr. mouse T2R5, tWGA-DsRed			*Zif268*	Sweet, bitter	Conventional	[206]
PBN	Mouse *SatB2*-Cre Vglut2-ihres-Cre	AAV1-Cre-GCaMP6s	AAV8-Cre-synaptophysin-mCherry AAV5-DIO-ChR2-EYFPin PBN	GCaMP6s	Ca^2+^ *c-fos*	Bitter	Miniaturized	[208]
Brain stem	Zebra fish	Tr. fish *Elav3*-GCaMP5		GCaMP5	Ca^2+^	Sweet, bitter, umami, sour	Two-photon	[213]
GC	Mouse	AVV1-GCaMP6s	AVV1-mCherry/microruby dextran (anterograde-thalamus)	GCaMP6s		Sweet, bitter, salt, sour		[192]
		AAV2/1-GCaMP6s	CAV2-Cre in hCAR x dTomato mice (retrograde-amygdala)			Bitter		[207]
	Mouse *T2R5/T1R2* knockout		AVV2-GFP (anterograde-thalamus)	GCaMP6s OGB-AM + sulforhodamine 101		Sweet, bitter, umami, salt		[202]

**Table 3 sensors-20-01811-t003:** Overview of recombinant systems that were used to study taste receptor function and new agonists/antagonists. Listed are different stimuli, which were investigated with Ca^2+^ sensors in host cells expressing recombinant proteins of the taste signaling machinery.

Host Cell	Stimuli	Ca^2+^ Sensor	Readout	Introduced Genes	Source
HEK293	Bitter	Fura-2	Microscopy	Gα15+T2R3-5-10-16	[47]
		Fura-AM		Gα16gust44/ Gα16gust37+T2R5-16	[259]
			Plate reader	Gα16gust44+T2R46	[288]
				Gα16gust44+T2R46-43-31	[265]
		Fluo-AM		Gα16gust44+T2R14	[289]
				Gα16gust44+variants of T2R16	[276]
				Gα16gust44+T2R43-44-4-46-50	[290]
				Gα16gust44+T2R31	[279]
				Gα16gust44+T2R16	[291]
				Gα16gust44+T2R43-44	[72]
				Gα15T2R16	[292]
		Fluo-4		Gα16gust44+hT2R31	[279]
				Gα16gust44+T2Rs (25 different types)	[73]
	Sweet	Fura-AM		V1R	[261]
		Fluo-AM		Gα15+T1R2+T1R3	[293]
				Gα16gust44+fT1R2+T1R3	[266,271]
		Fura-2		Gα16gust44+T1R2/R3 or T2R44	[294]
			Microscopy	Gα15+T1R2+T1R3	[83]
	Sweet, umami	Fluo-AM		Gα15+T1R2+T1R3 Gα15+T1R1+T1R3	[93]
	Acid	Fura-2-AM		PKD1L3+PKD2L1	[260]
					[264]
		Fura-2			[150]
		Fluo-AM			[154]
	Kokumi	Fluo-8, Flamindo (for cAMP)		CaSR	[263]

**Table 4 sensors-20-01811-t004:** Overview of reporter genes used to study taste signaling in specific cell populations. Different reporter genes were expressed under the control of diverse promoters of genes involved in the taste signaling cascade.

Promoter	Reporter Gene	Sensor	Readout		Stimuli	Source
Gustducin	lacZ β–galactosidase	No Ca^2+^ imaging	Ca^2+^	Microscopy	Bitter	[111]
	GFP	No Ca^2+^ imaging				[79]
		Fura-2				[141]
PLCβ2		CaOD				[297]
					KCl	[49]
PLCβ2, GAD					Sweet, bitter, umami, sour, salt	[114]
					Sweet, bitter, umami	[33]
					Sweet, bitter, umami, ACh	[42]
PLCβ2, GAD OXTR		Fura-2-AM			Oxytocin	[142]
T1R3 GAD					Glutamate	[311]
IP3R					Bitter, KCl	[50]
TRPM5 T1R3		Fura-2-AM			KCl	[113]
TRPM5	CFP	Fluo-5F			Ca^2+^	[112]
T2R32	GFP Sapphire	CaGD			KCl	[46]
PKD2L1	YFP	Carboxi-DFFDA + H+ uncaging	pH		Sour	[312]
		Fura-2-AM pHrodo-Red-AM	pH, Ca^2+^			[69]
GAD	GFP	No Ca^2+^ imaging	Ca^2+^			[67]
						[313]
		Fura-2				[150]
					IBMX-forskolin	[149]
PYY		No Ca^2+^ imaging			Bitter, lipids, sour, sweet, bitter, umami, salt	[314]

**Table 5 sensors-20-01811-t005:** Biosensor cells that can be used to study neurotransmitter release from taste cells. Upon sensing neurotransmitters, BC evoke intracellular Ca^2+^ transients that can be detected with Ca^2+^-sensor dyes. Abbreviations: 5HT: serotonin, NA: noradrenaline, Ach: achetylcholine.

BC	Ca^2+^ Sensor	Readout	Stimuli	Receptor	Neurotransmitter	Source
CHO	Fura-2-AM	Microscopy	KCl, sour, sweet, bitter, ATP	5HT_2c_	5HT	[36,328]
				5HT_2c_ or P2X_2_/P2X_3_	5HT ATP	[37,41,56,105,143]
			Adenosine			[325]
			Calcitonin gene-related peptide		NA 5-HT	[324]
			KCl, sour, sweet, bitter	5HT_2c_ or α_1A_ or dual	NA 5HT	[39]
			Sweet, bitter, KCl	P2X_2_/P2X_3_	ATP	[329]
			KCl, sour, sweet, bitter	GABA_B_+ Gαqo5	GABA	[40]
			Sweet, umami, bitter (fluorescein)	M3r P2X_2_/P2X_3_	ACh ATP	[42]
			KCl, taste mix, substance P	P GABA_B_+ Gαqo5 or 2X_2_/P2X_3_	GABA, ATP	[104]
COS-1	Fluo-4		Bitter, sour, depolarization, ACh, 5HT, NA, glutamate	P2Y endogenous	ATP ACh	[32,323]

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
