# Peer review of "Sensing Senses: Optical Biosensors to Study Gustation"

_sensors, 2020, doi:10.3390/s20071811_

Round 1

Reviewer 1 Report

The article presents a very comprehensive review on biosensor to study gustation. There are more than 300 references.

The tables are correct and the review is very well explained.

However, a very brief paragraph or section is missing to clarify where the references were obtained that are evaluated, the period of search or observation and percentage of references that were left out of the study.

Author Response

However, a very brief paragraph or section is missing to clarify where the references were obtained that are evaluated, the period of search or observation and percentage of references that were left out of the study.

Our response: At the end of the review, we have now included the criteria that we used for literature search and publications eligibility. As mentioned there (chapter 9, Literature research), two authors independently performed the search of relevant publications, based on open access sources. We tried to cover all relevant papers that make use of fluorescent optical biosensors in the context of taste: i.e sensors for calcium, sodium, voltage and pH. To our knowledge, these are all the functional readouts so far used in taste cells based on optical biosensors. We have searched for both, chemical dyes and genetically-encoded sensors. Regarding taste transduction, we have focused on the taste buds, but not on extra-oral tissue containing taste receptors. Regarding taste representation we have focused on the different brain regions involved in decoding information about the taste modalities, but not on brain regions involved in taste reward, satiety, motivation and learning. Although, we have tried to discuss all the publications found with the above-mentioned criteria, we have no clear evaluation of the number of missed articles.

Reviewer 2 Report

In this review, the authors first gave a brief, general review on taste-function and mechanism of action, and then systemically described the recent studies on applying either taste cell biosensors or cell line biosensors for the taste research. This article provides the basic and updated information to the investigators in the taste field to understand the contents and the related research. However, despite the compelling ideas, there are several issues of structure and wording that need to be considered.

major concern:

  1. The current manuscript is too long. Since the basic taste is discussed elsewhere, the 2nd part "taste-function and mechanism of action" can be removed and keep the contents precise.
  2. The definition of biosensors is ambiguous. Authors should make it clear that biosensor are used to detect taste cell function. Chemical dyes, genetically encoded indicators and the expression of reporter genes only make taste cells visible.
  3. GABA secretion by Type I cells reported in "Substance P as a putative efferent transmitter mediates GABAergic inhibition in mouse taste buds." should be cited in the review.
  4. Copyright of the figure 4 should be clarified in the figure legend.

minor concern:

  1. A typo in line436 should be corrected as "Roebber et al. (2019)".
  2. Petrosal ganglia should be mentioned in line488.
  3. [Huang 2011] in line839 should be corrected as [39].

Author Response

major concerns:

  1. The current manuscript is too long. Since the basic taste is discussed elsewhere, the 2nd part "taste-function and mechanism of action" can be removed and keep the contents precise.

Our response: The introductory part on the taste modalities was shortened by about 20 lines (now lines 62 to 281) and kept more compact, leaving only major findings and description of most recent publications. However, considering that: a) many readers may not be specialized in the field, b) the last comprehensive review describing all taste modalities is from 2017 (Roper et al. 2017, Reference 20), and since then some relevant papers have been published, c) the different signaling pathways are very complex and the open questions are not all reported in a single review , d) in the following chapters there are many references to the mechanisms described in the introductory part, we opted not to take it out completely. We hope, this is acceptable.

  1. The definition of biosensors is ambiguous. Authors should make it clear that biosensor are used to detect taste cell function. Chemical dyes, genetically encoded indicators and the expression of reporter genes only make taste cells visible.

Our response: As suggested, the definition of optical biosensors was further specified, see lines 49 to 60 of the revised manuscript.

  1. GABA secretion by Type I cells reported in "Substance P as a putative efferent transmitter mediates GABAergic inhibition in mouse taste buds." should be cited in the review.

Our response: This citation was added at lines 212f. and Table 5.

  1. Copyright of the figure 4 should be clarified in the figure legend.

Our response: Figure 4 shows unpublished data created by the authors.

minor concern:

A typo in line436 should be corrected as "Roebber et al. (2019)".

Petrosal ganglia should be mentioned in line 488.

[Huang 2011] in line839 should be corrected as [39].

Our response: Corrections were done as requested.

Reviewer 3 Report

Given that the paper is really complete and updated, it seems to me that the major part is matter of molecular and cellular biology more than optical sensing. And it is really a very long paper, but I cannot say how to divide it. For the rest, few mistype errors to correct and it can be published as it is.

Author Response

Given that the paper is really complete and updated, it seems to me that the major part is matter of molecular and cellular biology more than optical sensing. And it is really a very long paper, but I cannot say how to divide it. For the rest, few mistype errors to correct and it can be published as it is.

Our response: Three authors revised the text, hoping to have found and corrected all mistakes.

Reviewer 4 Report

This manuscript review on the applications and applied researches of biosensor cells expressing neurotransmitter receptors which are used to monitor substance release from taste cells.

This review is compact in thinking, strict in organization, rigorous in language and normative in reference. It systematically introduces relevant work to scholars in the same field or related fields. It has a certain summary and induction effect on the achievements in this field, and it is recommended to publish.

As I state in the submitted comments, this review is well-organized.

If there must be some suggestions for revision, I think the ABSTRACT of this review is too broad, which fails to highlight the key-points of the content of this review and states too much background.

Author Response

If there must be some suggestions for revision, I think the ABSTRACT of this review is too broad, which fails to highlight the key-points of the content of this review and states too much background.

Our response: Following the suggestions of the reviewer, the abstract was rewritten.